# Emotional disclosure as a therapeutic intervention in palliative care: a scoping review protocol

Daisy McInnerney,[1] Nuriye Kupeli,[1] Patrick Stone,[1] Kanthee Anantapong,[1,2] Justin Chan,[1] Bridget Candy[1]

¹Marie Curie Palliative Care Research Department, University College London, London, UK
²Department of Psychiatry, Faculty of Medicine, Prince of Songkla University, Hat Yai, Thailand

**Correspondence to**
Daisy McInnerney;
daisy.mcinnerney.18@ucl.ac.uk

## ABSTRACT

**Introduction** Emotional disclosure (ED) is a term used to describe the therapeutic expression of emotion. ED underlies a variety of therapies aimed at improving well-being for various populations, including people with palliative-stage disease and their family carers. Systematic reviews of ED-based psychotherapy have largely focused on expressive writing as a way of generating ED. However, heterogeneity in intervention format and outcome measures has made it difficult to analyse efficacy. There is also debate about the mechanisms proposed to explain the potential effects of ED.

We present a scoping review protocol to develop a taxonomy of ED-based interventions to identify and categorise the spectrum of interventions that could be classified under the umbrella term of 'emotional disclosure' in the palliative care setting. By mapping these to associated treatment objectives, outcome measures and explanatory frameworks, the review will inform future efforts to design and evaluate ED-based therapies in this population.

**Methods and analysis** The review will be guided by Arksey and O'Malley's five-stage scoping review framework and Levac's extension. The following electronic databases will be searched from database inception: CENTRAL, the Cumulative Index to Nursing and Allied Health Literature (CINAHL), PsycINFO, Scopus, Web of Science and MEDLINE. We will include peer-reviewed studies and reviews. We will also check grey literature, including clinical trial registers, conference proceedings and reference lists, as well as contacting researchers. Articles will be screened by at least two independent reviewers and data charted using an extraction form developed for this review. Results will be analysed thematically to create a taxonomy of interventions, outcome measures and theoretical frameworks.

**Ethics and dissemination** This review does not require ethical approval as it is a secondary analysis of pre-existing, published data. The results will inform future research in the development of ED-based interventions and evaluation of their efficacy in the palliative care setting. We will disseminate findings through peer-reviewed journals.

## Strengths and limitations of this study

► Despite the heterogeneity of the intervention repeatedly being cited as a limiting factor in evaluations of intervention efficacy, no previous study that we are aware of has systematically mapped the variety of emotional disclosure (ED)-based interventions, outcome measures and theoretical frameworks used in the palliative care setting.

► A rigorous, systematic approach will be applied to searching, screening, extracting and analysing the literature based on established scoping review methodological approaches.

► There may be challenges in identifying studies due to the differences in terms used to describe ED-based interventions.

► The review will be limited to studies of adult populations and published in the English language.

► The studies included in the review will not be appraised for methodological quality as this is outside the remit of scoping review methodology.

## INTRODUCTION

People living with a terminal illness often experience significant psychological, emotional and physical discomfort.[1 2] Family carers are also likely to experience psychological distress during and after supporting a relative through advanced illness.[2–4] Palliative care services aim to holistically address the physical, psychological and other needs of patients with advanced, life-limiting illness.[5] Psychotherapies form one important element of this palliative treatment approach.[6–8] However, access to such therapies in this setting is restricted by issues arising from limited availability of qualified professionals and evidence-based interventions well-adapted to the population.[9 10] Funding also poses a potential challenge, with a 2015 survey of hospices in the UK identifying significant concerns over freezing or reduction of statutory funding, and warning of its adverse effects on hospice services.[11 12] Furthermore, guidelines for the provision of psychological services in this population are limited. The National Institute of Health and Care Excellence last published guidance

on structuring psychological support in palliative care for adults in 2004 and only in cancer. It is not clear how widely these guidelines have been followed.[9 13] Indeed, a recent survey of clinical psychologists working in hospices in the UK indicated there is significant variability in how psychological support is provided.[9] Palliative care physicians also report limited access to psychological services for patients.[14] Taken together, this suggests that palliative care services might benefit from access to simple interventions that can be delivered by non-specialist healthcare practitioners, and with the potential for volunteer involvement. This could supplement the support offered by specialist practitioners to clinically distressed patients with complex needs.

Emotional disclosure (ED) is a term used to describe the therapeutic expression of emotions. This flexible therapy holds potential to be harnessed as a relatively low-cost, simple intervention in certain formats, such as expressive writing (EW).[15] Its flexibility also means it could be adapted to the specific needs of patients at a palliative stage of disease, for example, by modifying the method used to generate emotional expression (eg, typing or spoken disclosure). ED has long been a critical concept associated with the talk-based psychotherapies pioneered by Freud in the early 1900s.[15] However, rigorous research into the concept and associated therapies was relatively scant until the 1980s.[15] In 1986, Pennebaker and Beall introduced their influential EW intervention, which was designed to generate ED and led to a rapid expansion of research into this topic.[15 16] EW typically involves participants writing about their emotions associated with a traumatic experience for 15–20 min over 3–5 consecutive days. Since the introduction of Pennebaker and Beall's EW intervention, the format has been widely adapted to explore the boundary conditions of the intervention.[16–19] For example, tasks to induce ED have included writing about positive emotions or future goals, or spoken disclosure.[18–23] Poetry therapy is also recognised as an adaptation of the original EW paradigm.[22 24] Moreover, ED is recognised as a fundamental part of other forms of psychotherapy, such as music therapy[25] and art therapy.[26] In recent times, patients and family members are increasingly turning to blogging, social media and chatroom sites to disclose emotions around their experiences of their illness.[27–29]

The existence of numerous formats of ED-based therapies and behaviours complicates the process of exploring if and how such interventions might work. Yet, a clear understanding of the mechanisms linking cause and effect is considered fundamental to the development and study of complex interventions.[30 31] In the case of ED, there is unlikely to be a single underlying process, rather a framework of interacting mechanisms.[19 32 33] Processes that have been proposed to explain EW have been reviewed and include emotional inhibition, cognitive adaptation, exposure and emotional regulation,[19 32 33] yet no consensus has been reached on a unifying framework. Different methods of generating ED are also likely to

invoke different or overlapping processes. For instance, disclosure through EW may employ mechanisms related to language processing[34–36] and ED through art may function through sensory and motor processes.[37] Disclosure via online forums is likely to involve many of the mechanisms involved in social support.[38–40] Moreover, the therapeutic setting may influence which cognitive processes are initiated by ED. For instance, for patients at the palliative stage of disease, the potential effects of ED may be mediated by mechanisms related to meaning, control or closure in ways that may not occur in a healthy population.[41 42]

Mirroring the unclear processes underlying ED, the efficacy of ED-based therapies remains uncertain. In general populations, a meta-analysis of 146 studies identified a small but significant positive effect of ED-based interventions on both physical and psychological health outcomes in healthy populations.[43] It has been suggested that moderators, such as demographic, personality and existing emotional support, are also likely to influence the efficacy of ED-based therapies, highlighting the importance of targeting and tailoring such interventions.[43] In people receiving palliative care, evidence of efficacy also remains unclear. Much of the literature is focused on EW and reports mixed results. A recent systematic review and meta-analysis of four randomised controlled trials (RCTs) that examined EW in patients with advanced disease found it did not have a significant effect on any of the selected health-related outcomes.[44] However, the review also reported more promising results from studies that conducted linguistic analyses of EW in this population; these analyses identified that use of certain words, such as positive emotion words, was related to better emotional well-being.[45] They also found EW writers used more cognitive words associated with causal understanding, which the authors reported suggested cognitive changes.[46] Other related reviews have also uncovered mixed results. A systematic review of EW in patients with breast cancer (irrespective of stage) found the intervention reduced negative somatic symptoms at a 3 month follow-up, although it had no effect on psychological outcome measures.[47] Another review[48] assessing EW in a variety of cancers of all stages reported 6 of 13 studies as finding statistically significant, small to moderate benefits of EW on energy and sleep patterns,[49] depressive[50] and physical symptoms,[50 51] emotional support,[52] pain,[53] uptake of mental health services[54] and healthcare utilisation.[51] However, each of the 13 studies also reported some null effects, and the authors were unable to conduct a meta-analysis due to the heterogeneity of interventions and outcome measures used across the studies. A further review of EW in cancer populations found no evidence of EW efficacy on psychological, physical or quality of life outcomes,[55] while a review of therapeutic writing in patients with long-term illnesses also found no effects.[22] None of the studies of EW in populations with advanced disease reported significant negative effects, with the exception of Low et al (2010), who found women who

had been diagnosed with metastatic cancer for ≥4.7 years exhibited greater sleep disturbance following EW, whereas those more recently diagnosed did not.[56]

Such mixed results are also characteristic of other ED-based interventions in clinical populations. A preliminary scoping of the literature indicates that reviews of studies of explicitly ED-based interventions in the palliative care setting are lacking. However, individual studies have identified some benefits in this population. Kissane *et al* (2007) found supportive-expressive group therapy improved quality of life and treatment of depression in women with metastatic breast cancer.[57] Clements-Cortes found patients with a terminal illness were able to decrease depressive symptoms and social isolation and enhance relaxation by expressing their emotions through music therapy.[58] Indeed, a recent review of music therapy in patients receiving palliative care found it had a positive effect on pain, fatigue, anxiety and quality of life.[59] Given the heterogeneity of interventions included in the review, however, not all included interventions were necessarily designed specifically to evoke ED. Reviews of ED-based interventions in clinical populations with serious (although not explicitly palliative-stage) disease have identified mixed results. A systematic review of 52 trials reported that music interventions may have beneficial effects on anxiety, pain, fatigue and quality-of-life for people with cancer (of all stages), but that results were inconsistent across trials.[60] They also noted that, when asked, participants said they valued the opportunity for emotional expression and processing offered by the therapy. A further systematic review of creative psychological interventions (CPIs), which encompass the use of music, art, drama and dance/movement to express and process thoughts and emotions, found evidence of psychological but not physical benefits of CPIs across 10 trials in cancer patients.[61] Similarly, ED-based interventions in bereaved family members have also demonstrated mixed results,[62 63] although they have been less widely studied in this group than in patients.

While the efficacy of ED-based therapies in the palliative care setting remains uncertain, current reviews recommend further research to assess the true efficacy of each intervention.[22 44 47 48 55 60 61] This is due to a number of limitations of the current literature. First, current reviews are significantly limited by the heterogeneity of the format of interventions and outcome measures used across the studies they are reviewing.[22 44 47 48 55 60 61] Second, and tellingly, qualitative interviews show participants find certain interventions valuable, even where null effects are captured by quantitative measures.[48 60] This suggests current studies are not necessarily investigating the outcome measures that convey the benefit experienced by patients, which may be more abstract or existential in nature, particularly in patients with a palliative-stage disease.[48 64] Third, authors have noted a lack of effort to tailor interventions to the specific needs of people with advanced or terminal illness.[22 44 48] This could encompass, for example, offering audio recorded disclosure

as an alternative to written disclosure in EW studies, as some patients may lose the ability to write. Finally, methodological quality of studies included in the systematic reviews has been largely graded as low, with limitations due to sample size, methodological features, such as lack of randomisation and data reporting. In light of these significant limitations to existing research, future studies should aim to address these shortfalls. The broad nature of therapeutic ED, however, makes future research design challenging. This is due, in part, to the significant overlap of terms being used to describe various interventions, and a lack of clarity on the most appropriate format, outcome measures and underlying mechanisms.

To address these shortcomings, we plan to use scoping review methodology to conduct an explorative, yet systematic, investigation of the existing heterogeneous literature. Scoping review methodology can be used to clarify and map out complex concepts in a robust and replicable manner.[65–67] It is therefore a suitable method through which to identify, consolidate and categorise the existing literature into a taxonomy of ED-based interventions in the palliative care setting. Such a taxonomy will provide researchers with a framework to inform the design of future studies of ED-based interventions, by guiding selection of intervention format and outcome measures. Moreover, the taxonomy will map intervention efficacy, along with any reported facilitators and barriers, to intervention format, to help draw out potential mechanisms of action. If researchers use the taxonomy to inform study design, this should in turn lay the groundwork for more informative systematic reviews of ED-based interventions. Given the unique physical, psychological and emotional position (eg, in terms of needs and experiences) of patients at the palliative stage of disease and their family carers, the scope of this review will be limited to research conducted in the palliative care setting. However, these findings could also provide a springboard to help develop a taxonomy for ED-based interventions in other populations.

## Objectives
### Primary objective
To develop a taxonomy of ED-based interventions used in the palliative care setting, for people with advanced diseases and their family carers. The taxonomy will identify, categorise and define classes of intervention that fall under the umbrella term of 'emotional disclosure'.

### Secondary objective
To map classes of intervention defined in the taxonomy to (1) underlying mechanisms, (2) appropriate treatment objectives, (3) outcome measures, (4) any facilitators and barriers to intervention feasibility and (5) efficacy.

## METHODS AND ANALYSIS
This protocol is guided by the standard framework proposed by Arksey and O'Malley[66] and expanded by

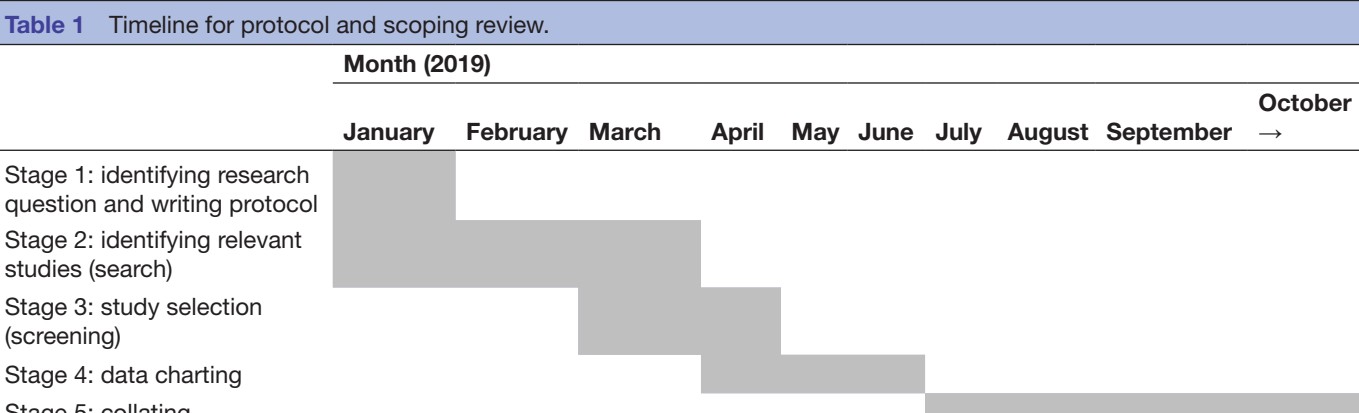

**Table 1** Timeline for protocol and scoping review.

| | Month (2019) | | | | | | | | | October → |
|---|---|---|---|---|---|---|---|---|---|---|
| | January | February | March | April | May | June | July | August | September | |
| Stage 1: identifying research question and writing protocol | ▓ | | | | | | | | | |
| Stage 2: identifying relevant studies (search) | ▓ | ▓ | ▓ | | | | | | | |
| Stage 3: study selection (screening) | | | | ▓ | ▓ | | | | | |
| Stage 4: data charting | | | | | ▓ | ▓ | ▓ | | | |
| Stage 5: collating, summarising and reporting results | | | | | | | | ▓ | ▓ | ▓ |
| Stage 6: consultation | *Throughout process at key stages (protocol development, screening and collating results)* | | | | | | | | | |

Cells shaded grey indicate the proposed timeline for completion of each stage.

Levac and colleagues[65] and the Joanna Briggs Institute.[67] These guidelines recommend organising the scoping review process into at least five stages, with an optional sixth stage:

► Stage 1: identifying the research question(s)—complete.
► Stage 2: identifying relevant studies.
► Stage 3: study selection.
► Stage 4: charting the data.
► Stage 5: collating, summarising and reporting the results.
► Stage 6: consultation.

The protocol has also been developed in line with scoping review best practice, as summarised in the completed Preferred Reporting Items for Systematic Reviews and Meta-Analyses for Protocols and Scoping Reviews included in online supplementary file 1 . Table 1 summarises the proposed timescale for the review.

### Stage 1: identifying the research question

To meet the objectives of the review, as outlined above, we will seek to thematically analyse insights from the following research questions:

► Which psychotherapeutic interventions delivered in patients at the palliative stage of disease and their family carers are categorised as, or explicitly grounded in, principles of ED? For example, what format are the interventions, how often are they delivered and by whom?
► What are the primary objectives of ED-based interventions delivered in this setting? For example, to enhance overall quality of life, physical or psychological health.
► What outcome measures are used to assess the efficacy of ED-based interventions in this setting?

► What theoretical frameworks are used to explain the mechanisms underlying ED-based interventions in this setting?
► What are the facilitators and barriers to ED-based intervention feasibility and efficacy in this setting?

### Stage 2: identifying relevant studies

The following inclusion criteria were developed in collaboration with key stakeholders, including physicians, psychologists and family carers involved in the provision of palliative care. Throughout the screening and data extraction process the criteria will be discussed within the research team and updated where necessary to ensure all relevant literature is being captured.[65]

► *Studies must use or make reference to a psychotherapeutic intervention that the authors state:*
  – involves 'emotional disclosure' or involves a task that requires participants to express or communicate feelings or emotions
  – as a core or critical element of the therapy
  – and that aims to improve some aspect of patient or carer well-being.
► *Articles published in the English language* (to prevent issues with intricacies of translation interfering with an effective definition of key terms).
► *The majority of the population of interest are adult participants (aged 18 and above):*
  – With a diagnosis of an advanced disease (eg, end-stage organ failure or advanced/metastatic/incurable cancers), and/or being explicitly treated with a palliative intent OR
  – Family carers of patients at a palliative stage of a disease.
  – Based on previous related reviews[44] that indicated that few studies meet these criteria, samples which

included >50% patients with advanced-stage disease will also be included.

▶ *All types of original research from within the peer-reviewed medical and nursing, psychological and social science literature will be included*, including RCTs, comparative studies (eg, non-randomised experiments, before-and-after studies), qualitative studies, case studies, ethnographies and diary studies.

▶ *Peer-reviewed conference abstracts* of papers not published in full will also be included if they are sufficiently detailed.

▶ *Review articles* that discuss ED as a psychotherapeutic intervention and make explicit mention of its use in the palliative care setting (or in patients with advanced disease and/or their family carers), including: systematic reviews, meta-analyses, meta-syntheses, scoping reviews, narrative reviews, rapid reviews, critical reviews and integrative reviews, opinion pieces, commentaries and editorial reviews.

### Exclusion criteria

The following resources will be excluded from data extraction and analysis:

▶ Studies with tasks that were not designed to be emotionally expressive, or which do not list ED (or similar) as a key feature of the intervention.

▶ Non peer-reviewed sources (eg, some book chapters and dissertations/theses); however, we will scan reference lists of relevant resources, and/or contact authors where appropriate.

No date limits will be applied to the searches, in order to capture the breadth of ED-based therapy delivery beyond the introduction of the well-cited EW paradigm in 1986.[16]

### Databases

The following electronic databases will be searched from database inception to March 2019: the CINAHL, Cochrane Central Register of Controlled Trials (CENTRAL), PsycINFO, Scopus, Web of Science and MEDLINE. We will also check the European Union Clinical Trials Register, clinicaltrials.gov, the European Association for Palliative Care and British Psychological Society (BPS) conference abstract proceedings for the last 7 (2012–2018) and 17 years (2001–2018), respectively. In addition, we will check the reference lists of relevant studies, review articles, book chapters and theses to identify further relevant citations. Finally, we will contact researchers who have expressed an interest in the field, via a research list compiled by the BPS, to ask if they are aware of any studies that may be relevant to this review. In case of uncertainty, authors of relevant studies will be approached to clarify whether studies meet the inclusion criteria for this review.

### Search strategy

The search strategy is based on an earlier systematic review exploring EW as a psychotherapeutic intervention in patients with advanced disease.[44] The search terms have been expanded to capture ED-based therapies more broadly, as well as including terms for advanced disease and palliative care. See online supplementary file 2 for an example search strategy used in the Ovid MEDLINE database that will be modified for each database, utilising keywords, MeSH terms and Boolean operators as appropriate. As per Levac and colleagues[65] recommendation for an iterative search strategy development process, we will review the search criteria throughout the screening process to update, expand or limit the search if required.

### Stage 3: study selection

The research team will meet to discuss preliminary inclusion and exclusion criteria during the protocol development phase. At least two reviewers will independently screen citations for inclusion to full article review stage. Reviewers will meet at the beginning, midpoint and final stages of the abstract review process to discuss challenges and uncertainties related to study selection, and to refine the search strategy and inclusion criteria if needed. Full article review will also be carried out independently by two researchers for articles which meet the inclusion criteria, or have unclear relevance during the screening phase. Where disagreements arise around inclusion, a third reviewer will be consulted to resolve disputes.

### Stage 4: charting the data

The research team will collectively develop the data-charting form based on the variables most relevant to the research questions. The form will be piloted using five articles, and the process and data-fields discussed between the research team prior to conducting the full data extraction procedure. Following full data extraction, the data from each independent reviewer will be compared and any discrepancies discussed to achieve consistency between reviewers.

A preliminary data extraction framework has been developed, tailored to answer each of the pre-defined research questions. Along with basic bibliographic information, information will be extracted about the study design, patient population, intervention characteristics, intervention objectives, outcome measures, underlying theoretical frameworks, intervention efficacy and proposed rationale for efficacy. A draft data-charting form for primary experimental studies is included in online supplementary file 3.

### Stage 5: collating, summarising and reporting the results

As per the guidelines of Levac and colleagues,[65] this stage will be conducted in three phases:

1. Analysis: to include both descriptive, quantitative analysis (eg, number of relevant studies within each intervention type; sample demographics) and qualitative thematic analysis (to explore how different ED-based interventions may be classified by format, objectives and/or other characteristics to inform the taxonomy).

2. Reporting the results of the analysis and producing the outcomes that refer to the study's research question(s).

3. Considering the meanings of the findings as they relate to the overall study purpose, and discussing the implications for future research, practice and/or policy.

Thematic analysis will be applied to understand the core, defining characteristics of each ED-based intervention.[68] From this analysis, we will work to develop a taxonomy of ED-based interventions. Thematic analysis of the objectives, outcomes and mechanisms will also be conducted, to enable us to map them onto intervention types. Exploratory analysis of facilitators, barriers and efficacy of specific interventions will also be conducted. We will examine whether there are any specific types of intervention that appear in studies using robust designs (eg, RCTs) to produce higher proportions of positive outcomes associated with specific outcome measures. We will also examine whether these patterns of efficacy are related to specific facilitators or barriers. The aim of this analysis will be to provide an indication of the reported efficacy and setting-specific requirements, with the intention of providing insights into the most useful direction for future work.

Results will be reported as tables, graphs and descriptive themes as appropriate. As well as reporting a taxonomy of ED-based interventions in the palliative care context, we will also discuss how it can be used to help guide future research into and implementation of ED-based psychotherapy in this setting.

## Stage 6: consultation

Although not mandated by the Arksey and O'Malley[66] framework, in the extension developed by Levac and colleagues[65] consultation with key stakeholders who may provide insights beyond the literature is essential. The research team who contributed to the development of this protocol includes a range of key stakeholders who will be engaged throughout the review process (including a palliative care consultant, a psychiatrist and researchers with expertise in EW, palliative care research and systematic review processes). An advisory group will also be consulted throughout the review process, including protocol development, results analysis and development of resulting conclusions and recommendations. The group includes health psychologist (Dr Nick Troop), clinical psychologist (Dr Penny Rapaport), former patient carer (Mr Peter Buckle) and evidence synthesis methodologist and palliative care nurse (Dr Kate Flemming).

## PATIENT AND PUBLIC INVOLVEMENT

As described in stage 6 (consultation), Mr Peter Buckle is a member of the advisory group. Peter has lived experience of caring for his wife throughout her terminal illness. He is a member of the Marie Curie Research Expert Voices Group, a group of volunteers with personal experiences of living with terminal illness who support Marie Curie's research activities. Peter's insights will be used throughout the review process, including development of the research protocol, results analysis and dissemination.

## ETHICS AND DISSEMINATION

We present a protocol for a comprehensive and rigorous scoping review of ED-based interventions used in patients and family carers in the palliative care setting. The results will be disseminated through traditional routes, including peer-reviewed journals, local and international conferences on palliative care and health psychology, and press releases, social media and blogs as appropriate. Through effective dissemination, the results of the review should help to inform more effective development, study and review of ED-based therapies in this patient population.

**Acknowledgements** The authors would like to thank all members of the advisory group for their thoughtful contributions to the study design: Mr Peter Buckle, Dr Kate Flemming, Dr Penny Rapaport and Dr Nick Troop.

**Contributions** DM contributed to the conception and design of the study, developed and tested the search strategy and inclusion criteria, and drafted the protocol. BC, NK, PS, KA and JC contributed to the conception and design of the protocol, specifically providing insight into the rationale, search strategy and inclusion criteria refinement, and critical review of the manuscript for clarity and intellectual content.

**Funding** The time spent on this review by author DM was supported by a Marie Curie and Economic and Social Research Council collaborative grant (grant number: ES/P000592/1). The time spent on the review by authors BC and PS comes from core funding of the research department by Marie Curie (grant number: MCCC-FPO-16-U) and an Alzheimer's Society (AS) funded fellowship awarded to NK (grant number: 399 AS-JF-17b-016). The time spent on the review by author KA comes from a grant provided by Faculty of Medicine, Prince of Songkla University, Thailand. The time spent on the review by author JC comes from a Marie Curie Chair's Grant (grant number: 509537). The work of all authors is also supported by the UCLH NIHR Biomedical Research Centre.

**Competing interests** None declared.

**Patient consent for publication** Not required.

**Ethics approval** Ethics approval is not required since the study involves only secondary analysis of data that has already been collected.

**Provenance and peer review** Not commissioned; externally peer reviewed.

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
