## [Reviewer comments · BMJ Open]

ARTICLE DETAILS

TITLE (PROVISIONAL)	Emotional disclosure as a therapeutic intervention in palliative care: a scoping review protocol
AUTHORS	McInnerney, Daisy; Kupeli, Nuriye; Stone, Patrick; Anantapong, Kanthee; Chan, Justin; Candy, Bridget

VERSION 1 – REVIEW

REVIEWER	Sean O'connor Ulster University
REVIEW RETURNED	07-Jun-2019

GENERAL COMMENTS	This is an interesting and very well written review protocol which aims to address a relevant research question. The introduction is comprehensive and well referenced providing a clear rationale for the proposed review and a detailed assessment of existing background evidence. The methods are appropriate and provided in sufficient detail, as is the planned synthesis of findings. Absence of methodological quality assessment is acceptable given the aims and design. An interesting aspect of the proposed review is the point made about the use of online methods of disclosure and the concept that these methods might be underpinned by social support. My only comments relate to the search strategy which is very broad and could well be very time consuming to complete. The authors acknowledge these challenges and include some measures to control for this but one possible suggestion that could reduce this complexity would have been to limit the dates of the primary searches.
--

REVIEWER	Adriana Coelho Nursing School of Coimbra (ESEnfC). Portugal
REVIEW RETURNED	08-Jun-2019

GENERAL COMMENTS	An interesting and timely issue in the field of palliative care, however, it lacks some improvements. The introduction section, in my view, is too extensive. However, there is fundamental information that should be included, and it does not appear. Like the information in the objectives section. In this sense, I suggest that the objective of revision be defined objectively and clearly. Information on the relevance and value of scoping should be included in the introduction. Finally, although the protocol has 29 pages, it does not have a temporary data extraction tool, it would be appropriate to present one in order to maximize the reader's understanding of what will be relevant to be extracted.
---

VERSION 1 – AUTHOR RESPONSE

REVIEWER 1 COMMENTS: Sean O'Connor, Ulster University:

"This is an interesting and very well written review protocol which aims to address a relevant research question. The introduction is comprehensive and well referenced providing a clear rationale for the proposed review and a detailed assessment of existing background evidence. The methods are appropriate and provided in sufficient detail, as is the planned synthesis of findings. Absence of methodological quality assessment is acceptable given the aims and design. An interesting aspect of the proposed review is the point made about the use of online methods of disclosure and the concept that these methods might be underpinned by social support."

- We would like to take this opportunity to thank Reviewer 1 for his positive comments. We agree that this is a pertinent topic for review, and recognise the importance of considering the implications of online adaptations to traditional disclosure methods.

"My only comments relate to the search strategy which is very broad and could well be very time consuming to complete. The authors acknowledge these challenges and include some measures to control for this but one possible suggestion that could reduce this complexity would have been to limit the dates of the primary searches."

- We acknowledge that the search terms are broad, resulting in a complex and time-consuming screening and analysis process. In the methods section, we explain that we will review our search criteria throughout the screening process. We have amended this description in response to Reviewer 1's comment to acknowledge that the search may also be limited if deemed appropriate. We have also taken steps to reduce the time taken to complete the review by recruiting a team of five researchers to work on screening and data extraction. We chose not to initially limit the search by date, as emotional-disclosure based interventions have been used for many decades and are a fundamental aspect of psychotherapy. As such, limiting by date may risk exclusion of seminal work in this field.

- We would also like to add, as outlined in the protocol introduction, that the format and terms used to describe emotional disclosure-based interventions are extremely heterogeneous, thus necessitating a broad search strategy. This is one of the key reasons underlying the need for this review. Our hope is that by capturing and categorising the broad scope of diverse interventions in a more formalised taxonomy, our review will facilitate design of more succinct search strategies for future reviews.

REVIEWER 2 COMMENTS: Adriana Coelho, Nursing School of Coimbra:

"An interesting and timely issue in the field of palliative care, however, it lacks some improvements. The introduction section, in my view, is too extensive."

- We would like to thank Reviewer 2 for her constructive feedback on the protocol. While we agree that the introduction is relatively long, we believe this is justified by the large body of pre-existing literature underlying the rationale for this review. A comprehensive overview of this literature is particularly important given the variable definitions and debate over the efficacy of emotional disclosure-based interventions. Furthermore, given the relevance of this review to a range of different readers, with likely differing areas of expertise (for example, researchers, clinical psychologists or psychotherapists as well as palliative care physicians), we feel it is important to provide a detailed overview of the topic area as a whole without assuming prior knowledge of core concepts.

"However, there is fundamental information that should be included, and it does not appear. Like the information in the objectives section. In this sense, I suggest that the objective of revision be defined

objectively and clearly. Information on the relevance and value of scoping should be included in the introduction."

- We have amended the objectives section to clarify the primary and secondary objectives of the review, and added a further research question in the methods section to clarify how we will fully meet the secondary objective. We have also moved the description of the relevance and value of the scoping review from the objectives and methods section to the end of the introduction section.

"Finally, although the protocol has 29 pages, it does not have a temporary data extraction tool, it would be appropriate to present one in order to maximize the reader's understanding of what will be relevant to be extracted."

-We have amended the methods section to direct readers to a draft data charting tool for primary experimental studies which is uploaded as Supplementary file 3.